# Multiscale Modelling Tool: Mathematical modelling of collective behaviour without the maths

**James A. R. Marshall** \***, Andreagiovanni Reina, Thomas Bose**

Department of Computer Science, University of Sheffield, Sheffield, United Kingdom

\* james.marshall@sheffield.ac.uk

**Data Availability Statement:** All software is available from GitHub (https://github.com/DiODeProject/MuMoT).

**Funding:** JARM acknowledges funding by the European Research Council (ERC - https://erc.

## Abstract

Collective behaviour is of fundamental importance in the life sciences, where it appears at levels of biological complexity from single cells to superorganisms, in demography and the social sciences, where it describes the behaviour of populations, and in the physical and engineering sciences, where it describes physical phenomena and can be used to design distributed systems. Reasoning about collective behaviour is inherently difficult, as the non-linear interactions between individuals give rise to complex emergent dynamics. Mathematical techniques have been developed to analyse systematically collective behaviour in such systems, yet these frequently require extensive formal training and technical ability to apply. Even for those with the requisite training and ability, analysis using these techniques can be laborious, time-consuming and error-prone. Together these difficulties raise a barrier-to-entry for practitioners wishing to analyse models of collective behaviour. However, rigorous modelling of collective behaviour is required to make progress in understanding and apply-ing it. Here we present an accessible tool which aims to automate the process of modelling and analysing collective behaviour, as far as possible. We focus our attention on the general class of systems described by reaction kinetics, involving interactions between components that change state as a result, as these are easily understood and extracted from data by nat-ural, physical and social scientists, and correspond to algorithms for component-level con-trollers in engineering applications. By providing simple automated access to advanced mathematical techniques from statistical physics, nonlinear dynamical systems analysis, and computational simulation, we hope to advance standards in modelling collective behav-iour. At the same time, by providing expert users with access to the results of automated analyses, sophisticated investigations that could take significant effort are substantially facil-itated. Our tool can be accessed online without installing software, uses a simple program-matic interface, and provides interactive graphical plots for users to develop understanding of their models.

europa.eu) under the European Union's Horizon 2020 research and innovation programme (grant agreement number 647704). The funders had no role in study design, data collection and analysis, decision to publish, or preparation of the manuscript.

# Introduction

Collective behaviour models, in which individuals interact and in doing so change state, describe a large variety of physical, biological, and social phenomena. One particularly general formulation is that of *reaction kinetics*, developed to describe the time evolution of chemical reactions, but also able to describe networks in molecular biology (*e.g.* [1]), collective behavioural phenomena such as decision-making in animal groups (*e.g.* [2]), demographic and ecological models such as predator-prey dynamics (*e.g.* [3]), epidemiological models (*e.g.* [3]), and social behaviour in human groups, such as opinion dynamics and economics (*e.g.* [4]). The generality of the reaction kinetics formalism is demonstrated by the fact that many of the aforementioned processes, although apparently quite different, are in fact described by the same dynamical equations; for example, the famous Lotka-Volterra equations were simultaneously developed in the description of a chemical reaction, and predator-prey dynamics [5, 6].

Modelling collective behaviour is essential to develop understanding, yet mathematical and computational modelling are skills than can be found in some disciplines much more than others. To understand commonalities and analogies across disciplines it would be beneficial to ensure a consistent standard of modelling is reached across all. However, it is unreasonable to expect all disciplines to ensure the same standard of mathematical training in their practitioners. Reaction kinetics have the advantage that they describe observations of a system in a very natural way, indeed the very way that experimental scientists tend to record those interactions. Reaction kinetics can also be transformed into mathematical equations according to a variety of procedures. The level of description attainable may vary, however. In their simplest form, mathematical models as Ordinary Differential Equations will assume infinitely large, well-mixed populations; this *mean-field* approach ignores fluctuations in subpopulation sizes due to the stochastic effects that small populations entail, and also ignores spatial heterogeneity and attendant sources of noise. Yet ODEs are analytically most tractable, and so enable general insights to be developed into the behaviour of an idealised version of the system of interest. By introducing finite population effects, noisy fluctuations around the mean-field solution can be studied; these can be approximated analytically, through the application of techniques from statistical mechanics, or numerically through efficient and probabilistically correct simulation of the Master Equation, which gives the continuous-time change in the probability density over the possible states of the system. These approaches are still idealisations, in that they ignore noise due to spatial effects, but they retain some tractability. Finally, one may analyse spatial sources of noise, by embedding a finite population in a spatial environment, such as a network, or a 2-dimensional plane or 3-dimensional volume. While in some cases analytic results may be possible, particularly in the case of networks, in general numerical simulation is required, sometimes referred to as Individual-Based Simulation or Agent-Based Simulation. This approach is therefore the most realistic, while also the least analytically tractable. In understanding the collective behaviour of some real-world system, therefore, the approach is generally to understand the simplest model of the system, then progressively introduce more realistic sources of noise in order to see if that behaviour is changed in important ways.

Taking all of the above points into consideration, we here present a Multiscale Modelling Tool, intended to simplify as much as possible the application of analytic and numerical techniques to descriptions of simple collective behaviour systems. The tool has the following objectives, and in the remainder of the paper we describe how these are achieved:

1. enable non-modellers to describe collective behaviour systems intuitively

2. enable a variety of analyses to be applied easily to such systems, accounting for increasingly realistic sources of noise

 a. infinite-population non-spatial noise-free dynamics

 b. non-spatial finite-population noisy dynamics

 c. spatial finite-population noisy dynamics

3. enable interactive exploration of analysis results

4. enable expert-level access to analysis results

5. minimise overheads for installation and use of the software

## Design and implementation

MuMoT (Multiscale Modelling Tool) is written in Python 3 [7] and designed to be run within Jupyter notebooks [8]. This enables MuMoT to be used in interactive notebook sessions using widgets, with explanations written in Markdown and $L^AT_EX$ to develop interactive computational documents, particularly suited to communication of results and concepts in research or teaching environments. A Jupyter notebook server can be deployed with a MuMoT installation to allow users to work through a standard web browser, without the need to install client-side software, facilitating access and uptake; at the time of writing, the interactive MuMoT user manual can be executed in this mode via Binder [9] (see [10]). Despite being primarily designed for interactive use, MuMoT uses a variant of the Model, View, Controller design pattern [11] enabling a separation between model descriptions, analytic tools applied to models, and interactive widgets for manipulation of analyses; this enables MuMoT to be used non-interactively, for example with routines called directly from user code.

 As MuMoT runs in Jupyter notebooks the user enters simple commands in notebook cells. Models are generated from intuitive textual descriptions, or from mathematical manipulation of previously-defined models, and most commands applicable to models result in interactive graphical output. To enable users to concentrate on presenting the key relevant concepts, users can partially or totally fix parameters in the resulting controllers, and have single controllers connected to multiple model views, with nesting of views if desired [10].

 MuMoT's implementation, testing, and documentation seeks to adhere to the best standards for scientific software deployment [12, 13].

## Specifying collective behaviour models

Users describe models as simple textual rules, standard in the description of reaction kinetics. We refer to individuals as *reactants* which can be, for example, different classes of individuals as in the case of chemical molecules or members of different biological species, or individuals having different changeable states as in the case of voter models, or robot swarms. Rules describe which reactants interact with each other, the resulting reactants, and the rate at which such reactions occur. For example, Fig 1 shows the description of a model of collective decision-making in honeybee swarms [2, 14] within MuMoT, and how this is parsed into a mathematical object.

 Models can also be created from the mathematical manipulation of other models; for example, it can be convenient to note that the frequency of one of the reactants can be determined from the frequencies of the remaining reactants, and the total system size, in any closed system where no reactant can be created or destroyed:

```
model2 = model1.substitute('U = N - A - B')
```

and to redefine rates in terms of other quantities, such as the qualities of potential nest sites in this example:

```
In [1]:   import mumot

          Created `%%model` as an alias for `%%latex`.

In [2]:   %%model
          $
          U -> A : g_A
          U -> B : g_B
          A -> U : a_A
          B -> U : a_B
          A + U -> A + A : r_A
          B + U -> B + B : r_B
          A + B -> A + U : s
          A + B -> B + U : s
          $
```

$U->A:g_A U->B:g_B A->U:a_A B->U:a_B A+U->A+A:r_A B+U->B+B:r_B A+B->A+U:s A+B->B+U:s$

```
In [3]:   model1 = mumot.parseModel(In[2])

In [4]:   model1.show()
```

$$U \xrightarrow{g_A} A$$

$$U \xrightarrow{g_B} B$$

$$A \xrightarrow{a_A} U$$

$$B \xrightarrow{a_B} U$$

$$A + U \xrightarrow{r_A} A + A$$

$$B + U \xrightarrow{r_B} B + B$$

$$A + B \xrightarrow{s} A + U$$

$$A + B \xrightarrow{s} B + U$$

**Fig 1. Specification of a collective behaviour model.** A model is described using simple textual rules denoting interactions between reactants in the system, and rates and which they are transformed into new combinations. Parsing this model description automatically results in a mathematical model object ready for analysis.

```
model3 = model2.substitute('a_A = 1/v_A, a_B = 1/v_B,
g_A = v_A, g_B = v_B, r_A = v_A, r_B = v_B')
```
or even in terms of the mean and difference between those qualities [2, 14]:
```
model4 = model3.substitute('v_A = \mu + \Delta/2, v_B = \mu -
\Delta/2')
```
Once parsed, a model exists as a mathematical object ready for analysis, as can be seen by asking to see the Ordinary Differential Equations (ODEs) that describe its time evolution:
```
model4.showODEs()
```
which results in the following system of equations:

$$
\begin{aligned}
\frac{\mathrm{d}A}{\mathrm{d}t} &:= -ABs + A\left(\frac{\Delta}{2} + \mu\right)(-A - B + N) - \frac{A}{\frac{\Delta}{2} + \mu} + \left(\frac{\Delta}{2} + \mu\right)(-A - B + N), \\
\frac{\mathrm{d}B}{\mathrm{d}t} &:= -ABs + B\left(-\frac{\Delta}{2} + \mu\right)(-A - B + N) - \frac{B}{-\frac{\Delta}{2} + \mu} + \left(-\frac{\Delta}{2} + \mu\right)(-A - B + N).
\end{aligned}
\tag{1}
$$

Eq 1 have been automatically derived from the rule-based description of the model we provided. Two techniques can be used to derive these ODEs, either a *mass action* heuristic similar to the one a mathematician would use to derive the ODEs, or a more involved statistical physics approach described in section 'Analysing noisy behaviour' (*e.g.* [15]). Both, however, have the same result.

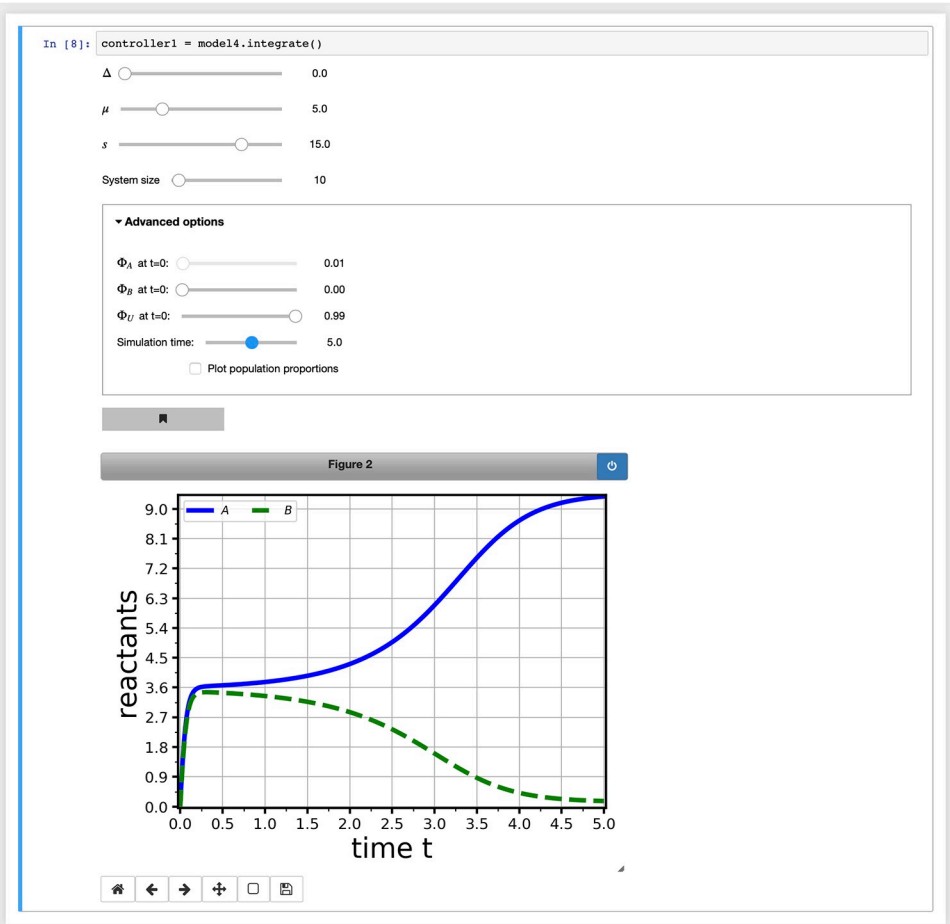

**Fig 2. Interactive manipulation of a model view via a controller.** Most model analysis commands result in an interactive graphical display of that analysis on the model. Users can explore and visualise the effects of changing free model parameters, and other analysis-specific parameters, through manipulating interactive controls.

Once a model has been parsed, a variety of analytic and numerical techniques can be applied to it. Many of these result in interactive graphical displays of the analysis, which users can manipulate to explore their model. For example, Fig 2 shows the result of performing a numerical integration on the model of Eq 1 within the notebook environment, using the `integrate()` command. Although not described in this paper, parameters can be fixed as desired to focus on a particular set of free parameters (*partial controllers*), and multiple views on the same model can be manipulated via a single controller (*multicontroller*). Users can also *bookmark* interesting parameter combinations to reproduce subsequently, and save the results from some views for analysis in external software packages. Such devices allow researchers and teachers to focus exploration and exposition of important concepts. Full details are given in the online user manual [10].

## Analysing noise-free infinite population behaviour

The most analytically tractable means of analysing collective behaviour are typically those that assume infinite populations; in this mean-field approach sources of intrinsic noise due to finite population effects are neglected, and space is also ignored. Thus understanding the noise-free

dynamics of a collective behaviour system is normally the most fruitful starting point in dealing with any new system.

**Numerical integration of ODEs and phase portraits.**   The simplest way to approach the noise-free dynamics of a system is often to integrate the ODEs that describe it. To achieve this MuMoT provides the `integrate()` method, which makes use of the `odeint` interface to numerical integrators implemented in Python's SciPy package `scipy.integrate` [16]. Solutions are displayed as interactive graphical output (see for example Fig 2). Plots can be presented either in terms of absolute numbers, or of population proportions (i.e. the number of 'particles' for each *reactant* divided by the system size at $t = 0$).

The dynamics of a MuMoT model can also be studied by means of a *phase plane* analysis. To visualise the model's trajectories in a *phase portrait* the methods `stream()` and `vector()` can be applied. Both methods depict phase planes representing the time evolution of the system as a function of its state; in a vector plot arrows give the direction in which the system will move, and their lengths show how fast, whereas in a stream plot lines show the average change of the system over time in finer resolution, and their shading represents the speed of change. It is also possible to calculate and display fixed points and noise around these; the corresponding theory and computations are introduced below. Stream plot examples are shown in Fig 4. More detailed explanations can be found in the online user manual [10].

**Bifurcations.**   Nonlinear dynamical systems may change behaviour qualitatively if model parameters are varied. To detect such transitions between different dynamic regimes MuMoT implements basic *bifurcation analysis* functionality by integrating with PyDSTool [17]. MuMoT's method enabling bifurcation analysis is called `bifurcation()`. Currently available is the detection of *branch points* (BPs) and *limit points* (LPs) of one-dimensional and two-dimensional systems; remember that a three-dimensional system may be reduced to a two-dimensional one using MuMoT's `substitute()` method. Detectable bifurcation points in MuMoT belong to the class of local codimension-one bifurcations. For example, BPs are observed for *pitchfork* bifurcations such as the one shown in Fig 5 (left panel). *Saddle-node* bifurcations are typical LPs (Fig 5 (middle and right panels)). For two-dimensional systems it may be desirable to directly compare the behaviour of both dynamical variables (or *state variables* as we call them within MuMoT) depending on a critical parameter in the same two-dimensional plot, where the bifurcation parameter is plotted on the horizontal axis. MuMoT allows users to plot single reactants as response variables, but also sums or differences of reactants, as illustrated in Fig 5 (left panel). For more information on the usage of `bifurcation()` we refer the reader to the online user manual [10].

When executing the `bifurcation()` method the following computations run behind the scenes. For a given parameter configuration, which includes the choice of the initial value of the bifurcation parameter, MuMoT attempts to determine all stationary states. If this is successful, MuMoT then starts the numerical continuation of each branch on which it found a stable fixed point. In case no stable fixed point could be detected, MuMoT numerically integrates the system using the initial conditions, and uses the final state at the end of the numerical integration as the starting point for the bifurcation analysis. If LPs or BPs were found those will be displayed and labelled in the bifurcation diagram. When MuMoT finds a BP it then tries to automatically start another continuation calculation along the other branch that meets the current branch at the BP. All curves that could be detected are displayed together at the end of the automated bifurcation analysis, colour-coded and shown with different line-styles to reflect the underlying stability properties of the corresponding stationary states. Fig 5 shows examples of different types of bifurcations that can be studied with MuMoT's `bifurcation()` method.

## Analysing noisy behaviour

Any real-world system is subject to noise, hence the next step in analysing a collective behaviour system is to examine deviations from the mean-field solutions of the model under such noise. There are two primary sources of noise, that due to finite population size, and that due to spatial distribution of the population; MuMoT enables analysis of both.

**Finite-population noise.** We start with intrinsic noise, due to finite population size. In any finite system the number of interactions fluctuates around an average value and hence so do the numbers of agents in the states available. The following derivation is based on the classical textbook by van Kampen [18]. In analogy to a typical chemical reaction let us consider a system of interacting agents $X_k$ with $k = 1, 2..., K$ being the different states agents might be in. Here $X$ denotes the type of agent and the state represented by index $k$ may be the commitment state. For example this could be a honeybee advertising a potential new nest site. The number of agents in state $k$ is denoted $n_k$; when agents interact the numbers in any state $k$ may change. Using integer stoichiometric coefficients denoted $\alpha_k$ and $\beta_k$ the change of the system's state following interactions may be described by

$$\alpha_1 X_1 + \alpha_2 X_2 + \cdots \rightarrow \beta_1 X_1 + \beta_2 X_2 + \cdots , \tag{2}$$

where the left-hand side characterises the state before the interaction (reaction) and the right-hand side the state after the interaction (reaction). All interaction processes are affected by the total number of agents. To account for this, we introduce the system size $\overline{V}$ as a formal (auxiliary) parameter that is necessary for the following derivation.

**The Master equation.** In order to sufficiently describe our system of interest, we need to compute the averaged macroscopic numbers and we also need to quantify the fluctuations around these averaged quantities. This may be achieved by means of the chemical Master equation, which can be written as follows [18]:

$$\begin{aligned}
\frac{\partial P(\{n_k\}; t)}{\partial t} &= \sum_i \left( r_+^{(i)} \overline{V} \left( \prod_k \mathbb{E}_k^{\alpha_k^{(i)} - \beta_k^{(i)}} - 1 \right) \prod_j \left( \frac{((n_j))^{\alpha_j^{(i)}}}{\overline{V}^{\alpha_j^{(i)}}} \right) P(\{n_k\}; t) \right. \\
&\quad \left. + r_-^{(i)} \overline{V} \left( \prod_k \mathbb{E}_k^{\beta_k^{(i)} - \alpha_k^{(i)}} - 1 \right) \prod_j \left( \frac{((n_j))^{\beta_j^{(i)}}}{\overline{V}^{\beta_j^{(i)}}} \right) P(\{n_k\}; t) \right),
\end{aligned} \tag{3}$$

where $\mathbb{E}$ is the step operator ([18], chapter VI, Eq 3.1), $\sum_i$ represents the sum over all reactions $i$, and rate superscripts $(i)$ denote the rates for reaction $i$. The first term in the sum on the right-hand side represents reactions as in Eq (2) (proportional to a constant interaction rate $r_+^{(i)}$) and the second term their inverse reactions (proportional to constant interaction rate $r_-^{(i)}$). Note that the inverse reaction does not always exist. If it exists, in a MuMoT model definition this would simply be written as an expression like the one in Eq (2), *i.e.* the convention used in MuMoT strictly follows Eq (2). For example, see input cell `In[2]` in Fig 1; there are also several examples in the online user manual to show how this works [10]. The expression $((n_j))^{\alpha_j} = n_j! / (n_j - \alpha_j)!$ is introduced as an abbreviation. Eq (3) describes the temporal evolution of the joint probability distribution that the system under study is in state $\{n_k\}$ at time $t$. Here, $\{n_k\}$ summarises all agents' individual states as a set. To express changes following interactions we make use of step operators $\mathbb{E}_k$ which increase or decrease the number of agents in state $k$ [18]. MuMoT automates the derivation of Eq (3) using the initial model definition according to Eq (2). The Master equation can be accessed as a symbolic equation object for further analysis by expert users, if so desired.

**van Kampen expansion of the Master equation.** In general, there are only very few examples for which Eq (3) can be solved exactly. In what follows we describe an approximation method known as *system size expansion* or *van Kampen expansion* that yields analytical expressions to approximate the solution of a Master equation. However, here we only introduce the main idea of the expansion method and refer to van Kampen's textbook [18] for further details. Let $\Phi_{X_k} = X_k / \overline{V}$ denote the proportion of the population $X_k$ given the system size $\overline{V}$. Note that $\Phi$ is a reserved symbol in MuMoT used to express population proportions—the analogue to concentrations of reactants in a chemical reaction. The probability to observe the system in state $n_k$ has a maximum around the macroscopic variable $\Phi_{X_k}$ with a deviation around that maximum of order $\sqrt{n_k} \sim \sqrt{\overline{V}}$ [18]. We may now replace the number $n_k$ by a new random variable, say $\eta_{X_k}$, according to [18]

$$n_k = \overline{V}\,\Phi_{X_k} + \sqrt{\overline{V}}\,\eta_{X_k}\,. \tag{4}$$

This also means that the probability distribution $P$ needs to be rewritten in the new variables, i.e. $P(\{n_k\};\,t) \to P(\{\eta_{X_k}\};\,t)$. Accordingly, the step operators $\mathbb{E}$ in Eq (3) are expanded to yield [18]

$$\mathbb{E} = 1 + \frac{1}{\sqrt{\overline{V}}}\frac{\partial}{\partial \eta_{X_k}} + \frac{1}{2}\frac{1}{\overline{V}}\frac{\partial^2}{\partial \eta_{X_k}^2} + \cdots\,. \tag{5}$$

Calculating the time derivative of $P(\{\eta_{X_k}\};\,t)$ by applying Eqs (4) and (5) to Eq (3) it is possible to get the equation for $P(\{\eta_{X_k}\};\,t)$ expressed in terms of different orders of the systems size $\overline{V}$ (note that the $\eta_{X_k}$ are time-dependent via $\Phi_{X_k}$ in Eq(4)). As a result, there are large terms $\propto \sqrt{\overline{V}}$ which should cancel, yielding the macroscopic equation of motion for $\Phi_{X_k}$. This corresponds to directly deriving the macroscopic ODE for $\Phi_{X_k}$ from the underlying reaction by applying the *law of mass action*. The next highest order in this expansion is $\propto \overline{V}^0$. Collecting all terms $\propto \overline{V}^0$ and neglecting all other terms ($\sim \mathcal{O}(\overline{V}^{-1/2})$) yields a Fokker-Planck equation with terms linear in $\eta_{X_k}$ (*linear noise approximation*). Although we do not attempt to solve Master equations or their approximations in the form of linear Fokker-Planck equations in MuMoT, we utilise the linear Fokker-Planck equation to compute analytical expressions that represent fluctuations and noise correlations, by deriving equations of motion for first and second order moments of $P(\{\eta_{X_k}\};\,t)$ according to

$$\begin{aligned}
\frac{\partial}{\partial t}\langle \eta_{X_i}\rangle(t) &= \int d\boldsymbol{\eta}\,\eta_{X_i}\frac{\partial}{\partial t}P(\{\eta_{X_k}\};\,t)\,, \\[6pt]
\frac{\partial}{\partial t}\langle \eta_{X_i}\eta_{X_j}\rangle(t) &= \int d\boldsymbol{\eta}\,\eta_{X_i}\eta_{X_j}\frac{\partial}{\partial t}P(\{\eta_{X_k}\};\,t)\,,
\end{aligned} \tag{6}$$

where $d\boldsymbol{\eta} = d\eta_{X_1}\cdots d\eta_{X_K}$, and $\partial P/\partial t$ represents the linear Fokker-Planck equation. Both van Kampen expansion and derivation of the linear Fokker-Planck equation can be readily performed in MuMoT. In addition, in MuMoT explicit expressions for first and second order moments following from Eq (6) may be derived. Furthermore, MuMoT can attempt to obtain analytical solutions for these equations in the stationary state.

All mathematical procedures concerning the Master equation and Fokker-Planck equation make extensive use of Python's SymPy package [19].

**Other methods to study noise in MuMoT.** Making use of MuMoT's functionality described in the previous paragraph, it is possible to compute and display the temporal

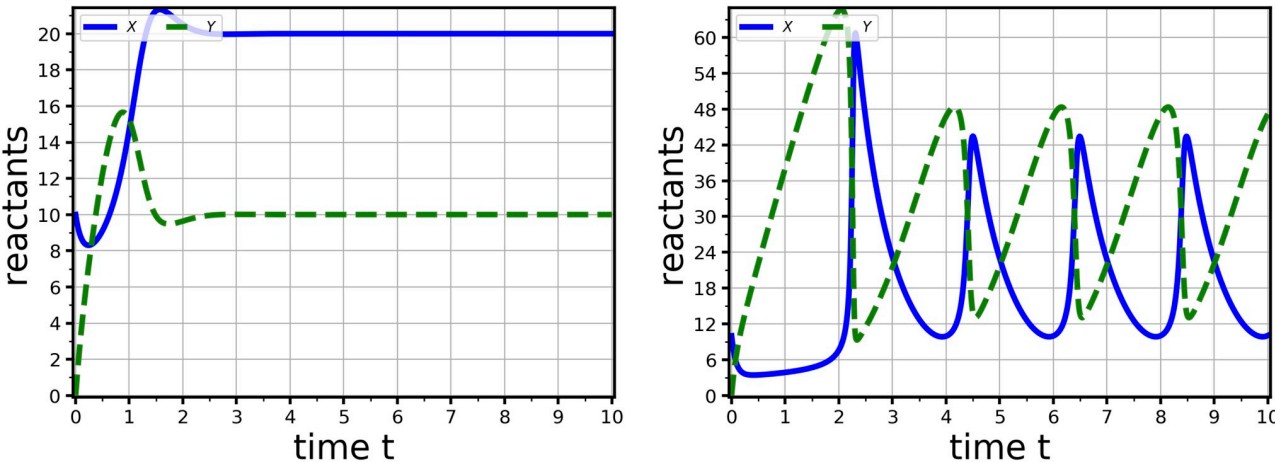

**Fig 3. Numerical integration of the Brusselator equations.** The Brusselator equations ([3], p.253) exhibit either stable (left) or oscillatory (right) dynamics according to the parameter values selected. Parameter sets: $\Phi_\alpha = \Phi_\beta = \chi = \delta = \gamma = \xi = 2.0$, $\Phi_{Xt(0)} = 1.0$, system size = 10 (left), $\Phi_\alpha = \chi = \delta = \gamma = \xi = 2.0$, $\Phi_\beta = 5.5$, $\Phi_{Xt(0)} = 1.0$, system size = 10 (right).

evolution of correlation functions $\langle \eta_{X_k}(t)\, \eta_{X_j}(0) \rangle$; examples of how to do this are given in the online user manual [10]. Noise can also be displayed in stream and vector plots; if requested then MuMoT tries to obtain the stationary solutions of the diagonal elements of the second order moments and then project these onto the direction of the eigenvectors of available stable fixed points of the macroscopic ODEs. If the system is too complicated and MuMoT cannot find an analytical solution, noise may be calculated by principled numerical simulation, as described below.

**Stochastic simulation.**   The Master equation of Eq (3) can be very difficult to solve for even very simple systems, therefore most studies resort to the complementary approach of numerical simulations [20]. Gillespie proposed a probabilistically exact algorithm for simulating chemical reactions called the *stochastic simulation algorithm* (SSA) [21]. Each simulation computes a stochastic temporal trajectory of the state variables from a given user-defined initial condition $\partial P(\{n_k\}; 0)$ for a user-defined maximum time $T$. Averaging various trajectories gives an approximation of the solution of Eq (3) (for a given $\partial P(\{n_k\}; 0)$) that increases in accuracy with the number of simulations. MuMoT implements the SSA via the command `SSA()`. The user can run a single simulation to generate a single temporal trajectory, or otherwise run several simulations and aggregate the data in a single plot. The user can visualise the entire temporal trajectory (in a plot similar to Fig 3), or the final population distribution $\partial P(\{n_k\}; T)$ in the form of either a barplot or as points in a 2-dimensional space plane (in which the two axes are state variables). Multiple trajectories can be aggregated in standardised ways of displaying probability distributions, *e.g.*, in the 2-dimensional space plane, simulation aggregates are visualised as ellipses centred on the distribution mean and with $1$-$\sigma$ covariance sizes (*e.g.* see the green ellipse in Fig 4(bottom panels)). This aggregate visualisation can be superimposed on to stream and vector field plots when requested, and if Eq (3) cannot be analytically solved by MuMoT, as discussed above.

**Spatial noise.**   MuMoT also enables the study of the effects of spatial noise on a model. Including spatial noise relaxes the sometimes simplistic assumption of a well-mixed system in which interactions between any group of reactants can always happen, at rates proportional to the product of their relative frequencies in the population. Instead, each reactant has a set of *available* reactants with which it can interact at each timestep. The set of possible interactions

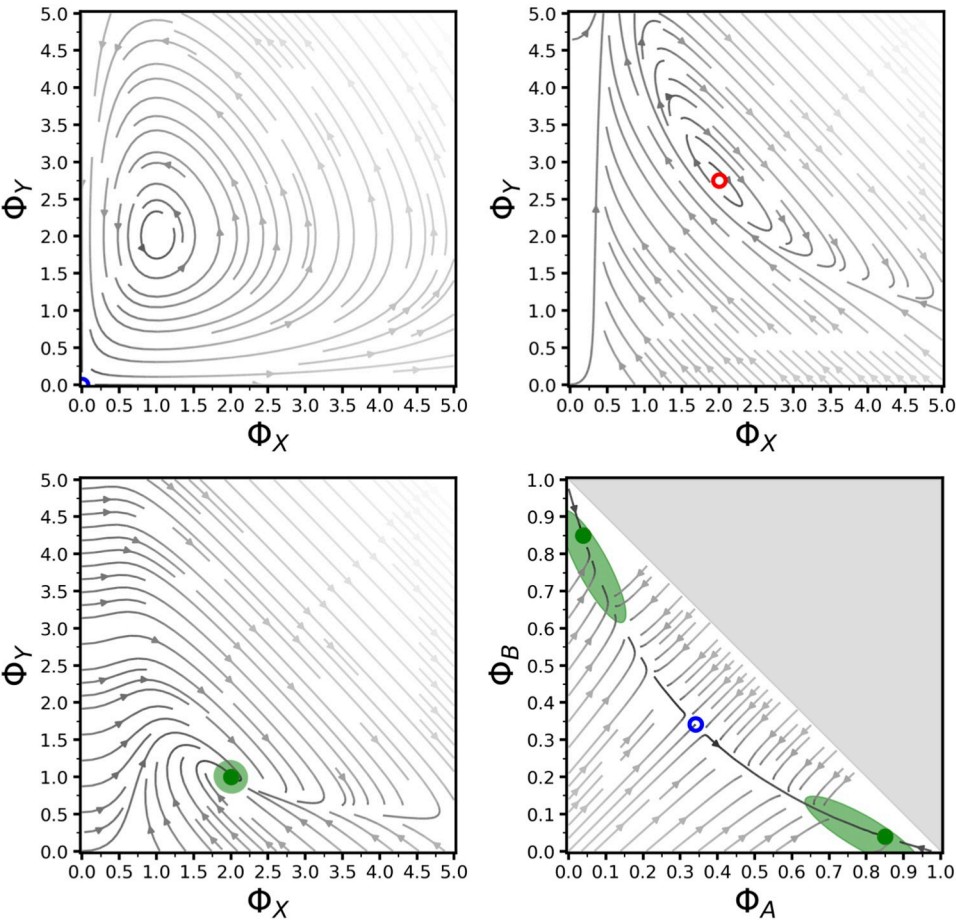

**Fig 4. Phase portraits with computed fixed points and noise.** Upper-left: oscillatory dynamics in the Lotka-Volterra equations ([3], p.79) (parameters $\Phi_A = \alpha = \beta = \gamma = 2.0$). Upper-right: limit cycle in the Brusellator ([3], p.253) (parameters $\Phi_\alpha = \chi = \delta = \gamma = \xi = 2.0$, $\Phi_\beta = 5.5$). Lower-left: global attractor with isotropic noise in the Brusellator ([3], p.253) (parameters $\Phi_\alpha = \Phi_\beta = \chi = \delta = \gamma = \xi = 2.0$, system size = 10). Lower-right: co-existence of two stable attractors in the honeybee swarming model [2], with anisotropic non-axis-parallel noise (parameters $\Delta = 0.0$, $\mu = 3.0$, $s = 10.0$, system size = 20, runs = 100). Line shading indicates speed of flow, with darker representing faster. Fixed points are denoted as stable (dark solid green circle), saddle (hollow blue circle), or unstable (hollow red circle). Light green ellipses represent $1$-$\sigma$ noise around stable fixed points.

corresponds to the system's interaction topology, which the user can select among a set of standard graph structures. Graphs are handled by MuMoT through the functionalities offered by the NetworkX library [22] which allows advanced users to easily add new topologies. In the first MuMoT release, the available topologies are the complete graph, the Erdös–Rényi random graph [23], the Barabási–Albert scale-free network [24], and the random geometric graph [25]. The latter is constructed by locating at random uniform locations the reactants in a square environment with edge length 1, and allowing interaction between two reactants when their Euclidean distance is less than or equal to a user-defined distance. The topology of the random geometric graphs can be static or time-varying. The latter is implemented by letting each reactant perform a correlated random walk in the 2-dimensional environment and recomputing the topology each time based on the new distances between reactants.

Spatial noise is difficult to compute analytically in an automatised way, therefore MuMoT computes it numerically via individual-based simulations. Each reactant is simulated as an agent which probabilistically interacts at synchronous discrete timesteps with the available

reactants. The agent's behaviour is automatically implemented from the model's reaction kinetics as a probabilistic finite state machine following the technique proposed in [26]. Along with the agents' behaviour, MuMoT automatically sizes the timestep length to match the time-scale with the population-level descriptions (*e.g.* ODEs and Master equation). This feature can be particularly convenient if the user aims at a quantitative comparison between model description levels. Similarly to SSA simulations, the user can select to run individual simula-tions or to aggregate results from multiple independent simulations to compute statistical distributions.

## Results

All results can be reproduced using the `MuMoTpaperResults.ipynb` Jupyter notebook [10].

### Numerical integration

To illustrate the numerical integration functionality of MuMoT we repeat analyses of the Brus-selator equations ([3], p.253) in Fig 3. The equations have two dynamical regimes, one with a single globally stable attractor when $\Phi_\beta \leq \Phi_\alpha^2$ (Fig 3 (left)), and one in which a stable limit cycle exists when $\Phi_\beta > \Phi_\alpha^2$ (Fig 3 (right)).

### Phase portraits with fixed point and noise calculations

We illustrate the phase portrait functionality of MuMoT in Fig 4 by repeating analyses of a variety of equation systems: the classical Lotka-Volterra equations ([3], p.79), the Brusellator equations ([3], p.253), and a model of collective decision-making by swarming honeybees [2, 14]. These systems can exhibit a variety of dynamics including: oscillations (Fig 4 (upper-left)), unstable fixed points with limit cycles (Fig 4 (upper-right)), globally stable attractors (Fig 4 (bottom-left)), and stable attractors co-existing with saddle points (Fig 4 (bottom-right)). When stable fixed points are present MuMoT can calculate or compute the equilibrium noise around them, dependent on system size (Fig 4 (bottom)); this can be either isotropic (Fig 4 (bottom-left)), or anisoptropic and/or non-axis-parallel (Fig 4 (bottom-right)). This latter case is particularly interesting because the correct noise around the fixed point may differ substan-tially from simply adding Gaussian noise to the dynamical equations.

### Bifurcation analysis

MuMoT's bifurcation analysis functionality is illustrated through reproducing a number of bifurcation analyses [14] of the honeybee model presented above [2] (Fig 5). These reveal con-ditions under which the dynamics exhibit: (i) a pitchfork bifurcation (Fig 5 (left)), a sample post-bifurcation phase portrait for which is presented in Fig 4, (ii) an unfolding of the pitch-fork bifurcation (*i.e.* saddle-node bifurcation) (Fig 5 (centre)), and (iii) a hysteresis loop (Fig 5 (right)). These can be compared to figures 5(i)-(iii) of [14].

### Finite population and spatial numerical simulation

MuMoT can be used to perform a variety of spatial numerical simulations, illustrated in Fig 6 for the honeybee swarming model introduced above [2, 14]. Non-spatial finite-population simulation reproduces the statistics of deadlock breaking observed in [14] (Fig 6 (left)). Spatial noise can also be incorporated either by embedding the model in a network (Fig 6 (centre)) or 2d-plane (Fig 6 (right)).

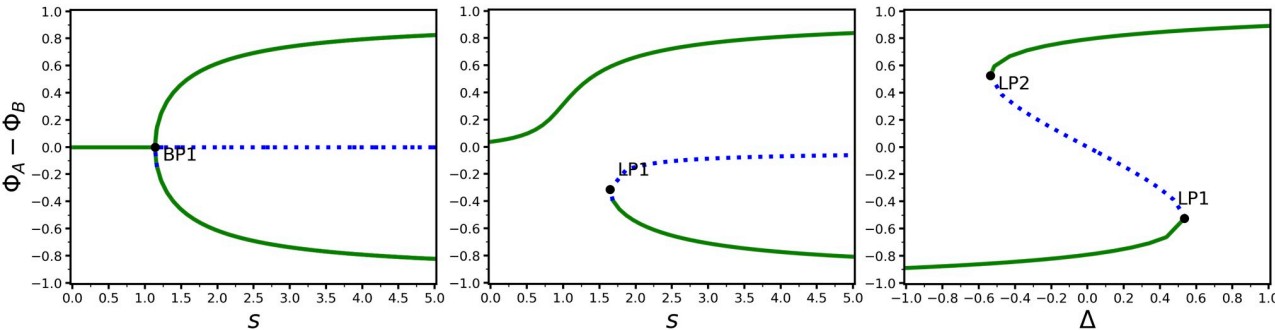

**Fig 5. Bifurcation analysis of a nonlinear decision-making model.** Bifurcations of the honeybee swarming model [2, 14]. Left: symmetry breaking in the two decision populations through a pitchfork bifurcation, with strength of cross-inhibitory stop-signalling $s$ as the bifurcation parameter (cf. [14] Fig 5i) (parameters $\Delta = 0.0$, $\mu = 4.0$). Centre: unfolding of the pitchfork bifurcation into a saddle-node bifurcation (cf. [14] Fig 5ii) (parameters $\Delta = 0.1$, $\mu = 4.0$). Right: hysteresis loop with option quality difference $\Delta$ as the bifurcation parameter (cf. [14] Fig 5iii) (parameters $\mu = s = 4.0$). Solid black lines denote stable branches, dashed blue lines denote unstable branches.

## Derivation of the Master equation and expansion to derive the Fokker-Planck equation

Here we reproduce the analysis presented in [18] (pp. 244-246) to derive the Master Equation and Fokker-Planck equation for the following toy model:

$$
\begin{aligned}
(A) &\xrightarrow{k} X \\
X + X &\xrightarrow{h} \emptyset + \emptyset
\end{aligned}
\tag{7}
$$

The automated analysis results in

$$
\frac{\partial}{\partial t} P(X, t) := \frac{Ak}{\overline{V}} (\mathrm{E}_{\mathrm{op}}(X, -1) - 1)\overline{V} P(X, t) + h(\mathrm{E}_{\mathrm{op}}(X, 2) - 1)\frac{X}{\overline{V}}(X - 1)P(X, t)
\tag{8}
$$

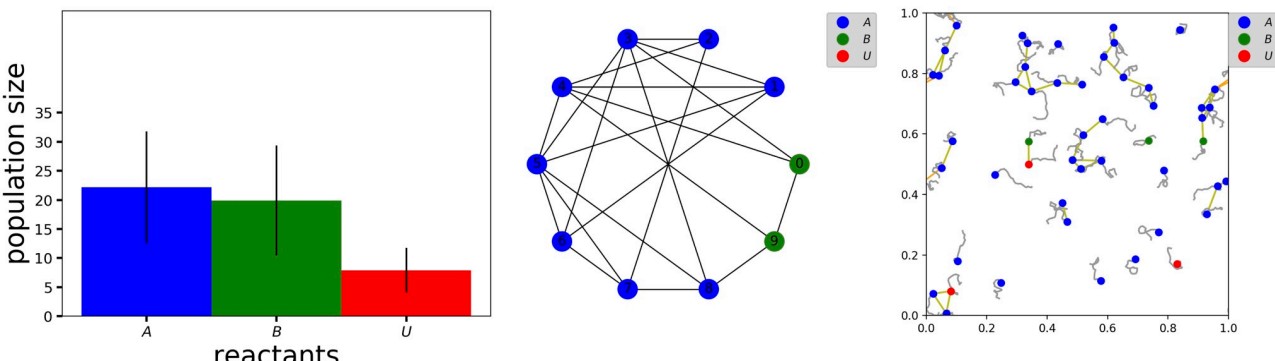

**Fig 6. Numerical simulations of a nonlinear decision-making model.** Numerical simulations of the honeybee swarming model [2, 14] given various sources of noise. Left: finite-population noise effects during symmetry-breaking in a well-mixed model (cf. [14] Movie S1) (parameters $\Delta = 0$, $\mu = 3.0$, $s = 3.0$, $\Phi_{Ut(0)} = 1.0$, system size = 50, time = 10, runs = 10). Centre: finite-population and spatial noise effects due to embedding the model in a random graph. Right: finite-population and spatial noise effects due to embedding the model in a plane, with agents performing correlated random walks; traces indicate recent agent paths, links indicate current interaction events.

and

$$\frac{\partial}{\partial t} P(\eta_X, t) := \frac{\Phi_A k}{2} \frac{\partial^2}{\partial \eta_X^2} P(\eta_X, t) + 2\Phi_X^2 h \frac{\partial^2}{\partial \eta_X^2} P(\eta_X, t) + 4\Phi_X \eta_X h \frac{\partial}{\partial \eta_X} P(\eta_X, t) + 4\Phi_X h P(\eta_X, t) \quad (9)$$

as expected.

A substantially more complicated example derivation, for the honeybee swarming model of Eq 1 [2, 14], is presented in S1 Text. This derivation is equivalent to that performed in [2] and results in the same dynamical equations.

## Availability and future directions

MuMoT is available as source code, as a package for Python 3 [27] via PyPI (`pypi.python.org`), and as a server-based installation currently exemplified by free-to-use access to the interactive user manual and other notebooks using the Binder service [9], which requires only a web browser to use. MuMoT is written in Python 3 and integrates with Jupyter Notebooks [8] and as such is platform-independent. Non-interactive aspects of MuMoT's functionality can also be accessed through using it as a standalone Python package, enabling its modelling and analysis functionality to be used from within third-party code projects. MuMoT is available under the GPL licence version 3.0, and makes use of other software available under the MIT licence. For further details including links to usage information are available at `github.com/DiODeProject/MuMoT/`.

Numerous software products have been proposed to perform subsets of the analyses offered by MuMoT. For instance, several tools offer the possibility to run the SSA and efficiently analyse reaction kinetics models [28–36]. Similarly, software to analyse mean-field dynamical systems and perform bifurcation analysis is widely available, *e.g.* MATCONT for Matlab [37], or the Dynamica package for Wolfram Mathematica [38]. Linear noise approximations have previously been implemented as well [32]. Several tools offers software to simulate complex systems, dynamical networks, and agent-based models [39–41], some of which run as Jupyter notebooks as MuMoT does [42, 43].

In contrast to the previous solutions, MuMoT combines ease-of-use with a multi-level analysis that spans from ODEs analysis, to statistical physics approximations, bifurcation analysis, and SSA and multiagent simulations, integrated within a simple interactive notebook document interface. This makes MuMoT particularly appropriate for non-experts to learn to build models and apply complex mathematical and computational techniques to them, to communicate research results, and to enable students to interactively explore models, and modelling and analysis techniques.

Future work should focus on integrating MuMoT with other software and standard. For example, the simple textual input method for MuMoT models is very accessible to non-experts, but precludes more sophisticated use cases. Import and export via interchange formats such as Systems Biology Markup Language (SBML) [44] would enable users to connect between MuMoT for general analysis, and external specialist software packages for more detailed analyses; for example StochSS [36] can run the SSA algorithm on cloud infrastructure for larger-scale computations, and perform parameter sweeps and estimation. Embracing data interchange formats will allow MuMoT to take its place as an integral part of the growing ecosystem of open-source modelling software.

## Supporting information

**S1 Text. Sample model analyses.** van Kampen expansion and other analyses of the stop-signal model (Eq 1).
(PDF)

## Acknowledgments

We thank Will Furnass with assistance in deployment, documentation, and continuous integration. We thank Renato Pagliara for assistance in testing on the Windows platform.

## Author Contributions

**Conceptualization:** James A. R. Marshall.

**Formal analysis:** Thomas Bose.

**Funding acquisition:** James A. R. Marshall.

**Methodology:** James A. R. Marshall, Andreagiovanni Reina, Thomas Bose.

**Project administration:** James A. R. Marshall.

**Software:** James A. R. Marshall, Andreagiovanni Reina, Thomas Bose.

**Validation:** James A. R. Marshall, Andreagiovanni Reina, Thomas Bose.

**Writing – original draft:** James A. R. Marshall, Andreagiovanni Reina, Thomas Bose.

**Writing – review & editing:** James A. R. Marshall, Andreagiovanni Reina, Thomas Bose.

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
