## [Decision Letter · Decision Letter 0]

2 Sep 2019

[EXSCINDED]

PONE-D-19-18541

Multiscale Modelling Tool: Mathematical Modelling of Collective Behaviour without the Maths

PLOS ONE

Dear Prof Marshall,

Thank you for submitting your manuscript to PLOS ONE. After careful consideration, we feel that it has merit but does not fully meet PLOS ONE’s publication criteria as it currently stands. Therefore, we invite you to submit a revised version of the manuscript that addresses the points raised during the review process, particularly to clarify the master equation, make notations more consistent with the literature in the domain of probability and clarify notations as requested by Referee 1. 

We would appreciate receiving your revised manuscript by Sep 29 2019 11:59PM. To enhance the reproducibility of your results, we recommend that if applicable you deposit your laboratory protocols in protocols.io, where a protocol can be assigned its own identifier (DOI) such that it can be cited independently in the future. For instructions see: http://journals.plos.org/plosone/s/submission-guidelines#loc-laboratory-protocols

We look forward to receiving your revised manuscript.

Kind regards,

Jonathan David Touboul

Academic Editor

PLOS ONE

Reviewers' comments:

Reviewer's Responses to Questions

**Comments to the Author**

1. Is the manuscript technically sound, and do the data support the conclusions?

Reviewer #1: Yes

Reviewer #2: Yes

2. Has the statistical analysis been performed appropriately and rigorously? 

Reviewer #1: N/A

Reviewer #2: N/A

3. Have the authors made all data underlying the findings in their manuscript fully available?

Reviewer #1: Yes

Reviewer #2: Yes

4. Is the manuscript presented in an intelligible fashion and written in standard English?

Reviewer #1: Yes

Reviewer #2: Yes

5. Review Comments to the Author

Reviewer #1: Its a nice well-written article. As the authors state, there is a need for accessible software for non-specialists to study chemical reaction networks, so this is very timely.

I have the following comment on the Master Equation on page 7 I am not familiar with your way of writing the Master Equation in (3). Is there a specific equation in Van Kampen that you are citing? Maybe you should indicate this. Normally $\\mathbb{E}$ means expectation, so it would be better to use another notation for the step operator.

Also it is not quite clear what the superscripts of $\\mathbb{E}$ are indicating.

In general, a few more details are needed here.

Reviewer #2: This article is a presentation of a computational tool called MuMoT, which aims to automate the analysis and formalize the process of modeling collective behaviour. Using the learning by examples technique, authors are able to present the particularities of the software and highlight how it can be used to perform description of collective behaviour systems in the cases with and without noise. This perspective responds to the three levels of description that a modeler can use when attacking a general problem micro, meso and macroscopic viewpoints.

The software presented is very interesting. Written on Python 3, inherits all the good properties of that computational language in particular, it is easy to use and has a very fast learning curve. Despite the fact that one could ask for a statistical sub-routine ubiquitous to experimentalists and sampling, MuMoT is a great tool and surely will help non-modellers to describe and understand their intuitions. The problem solved by MuMoT is not new, and it is not the first piece of software solving it, however the novelty of the work is that combines techniques from different mathematical areas in a easy-to-use manner.

I recommend publishing the paper in PLOS ONE.

6. PLOS authors have the option to publish the peer review history of their article (what does this mean?). If published, this will include your full peer review and any attached files.

Reviewer #1: No

Reviewer #2: No

---

## [Author Response · Author response to Decision Letter 0]

9 Sep 2019

I have the following comment on the Master Equation on page 7 I am not familiar with your way of writing the Master Equation in (3). Is there a specific equation in Van Kampen that you are citing? Maybe you should indicate this. Normally $\\mathbb{E}$ means expectation, so it would be better to use another notation for the step operator.

Also it is not quite clear what the superscripts of $\\mathbb{E}$ are indicating.

In general, a few more details are needed here.

Thank you. The notation used is that used in the third edition of van Kampen’s textbook; we have added a citation to the precise definition in said textbook, and clarified the use of superscripts as requested. We have additionally changed to use the bbold package to improve the similarity to the step operator as typeset in van Kampen.

---

## [Editor Report · Decision Letter 1]

11 Sep 2019

Multiscale Modelling Tool: Mathematical Modelling of Collective Behaviour without the Maths

PONE-D-19-18541R1

Dear Dr. Marshall,

We are pleased to inform you that your manuscript has been judged scientifically suitable for publication and will be formally accepted for publication once it complies with all outstanding technical requirements.

With kind regards,

Jonathan David Touboul

Academic Editor

PLOS ONE
---

## [Editor Report · Acceptance letter]

16 Sep 2019

PONE-D-19-18541R1 

Multiscale Modelling Tool: Mathematical Modelling of Collective Behaviour without the Maths 

Dear Dr. Marshall:

I am pleased to inform you that your manuscript has been deemed suitable for publication in PLOS ONE. Congratulations! Your manuscript is now with our production department. 

With kind regards,

on behalf of

Dr. Jonathan David Touboul 

Academic Editor

PLOS ONE